

# Impact of livestock grazing management on carbon stocks: a case study in sparse elm woodlands of semi-arid lands

Yi Tang

School of Emergency Management, Institute of Disaster Prevention, Sanhe, Hebei Province, China

## ABSTRACT

Livestock grazing is a widespread practice in human activities worldwide. However, the effects of livestock grazing management on vegetation carbon storage have not been thoroughly evaluated. In this study, we used the system dynamic approach to simulate the effects of different livestock grazing management strategies on carbon stock in sparse elm woodlands. The livestock grazing management strategies included rotational grazing every 5 years (RG5), prohibited grazing (PG), seasonal prohibited grazing (SPG), and continuous grazing (CG). We evaluated the carbon sequestration rate in vegetation using logistical models. The results showed that the carbon stock of elm trees in sparse woodlands was 5–15 M g ha$^{-1}$. The values of the carbon sequestration rate were 0.15, 0.13, 0.13, and 0.09 Mg C ha$^{-1}$ year$^{-1}$ in RG5, PG, CG, and SPG management, respectively. This indicates that rotational grazing management might be the optimal choice for improving vegetation carbon accumulation in sparse woodlands. This study contributes to decision-making on how to choose livestock grazing management to maintain higher carbon storage.

# INTRODUCTION

Global warming, which is primarily caused by excessive emissions of CO$_2$, has attracted worldwide attention (*Lv et al., 2019*). One important strategy for reducing anthropogenic CO$_2$ emissions is carbon storage (*Min et al., 2018*). The terrestrial biosphere, a repository of approximately 2,000 Gt of carbon, plays a crucial role in carbon storage (*Ruhland & Niere, 2019*). Vegetation, as a major component of the terrestrial biosphere, is responsible for a large proportion of carbon storage (*Hou et al., 2015*; *Fu et al., 2019*). Trees, as the largest and longest-lived plants, are particularly important for carbon storage (*Zellweger et al., 2022*). Vegetation plays a pivotal role in regulating the carbon cycle and storing carbon primarily by absorbing CO$_2$ through photosynthesis and subsequently releasing some of it back into the atmosphere through respiration (*Qiu et al., 2020*). Therefore, strategies that enhance vegetation carbon storage can effectively mitigate global warming (*Iglesias, Barchuk & Grilli, 2012*; *Piao et al., 2018*).

Arid and semi-arid areas together constitute almost one-third of the world's land area (*Zhang et al., 2018*). In these regions, livestock grazing is a prominent human activity. It is

Corresponding author
Yi Tang, tangyi@lnu.edu.cn

considered as an effective approach for regulating carbon storage in soil and vegetation (*Chen et al., 2012*; *Wu et al., 2014*). The effects of grazing exclusion and its duration on carbon storage in soil and vegetation have been studied in previous studies (*Medina-Roldan, Paz-Ferreiro & Bardgett, 2012*; *Xiong et al., 2016*; *Wang et al., 2018*). For example, the effects of fencing on carbon stocks were explored in degraded alpine grasslands (*Li et al., 2013*), and the duration of exclosure practice was investigated in a degraded grassland in China (*Li et al., 2012*).

Grazing management plays a pivotal role in shaping the carbon dynamics of terrestrial ecosystems (*Polley et al., 2008*). At the most fundamental level, grazing alters plant biomass, which directly impacts the amount of carbon that vegetation can sequester from the atmosphere (*Farrell et al., 2015*). For instance, moderate grazing can stimulate plant growth due to the removal of old plant material, which can pave the way for new shoots. This often results in a net increase in photosynthetic activity and, thus, increased carbon capture. In contrast, the balance of different plant species, dictated in part by grazing intensity, can also influence the carbon storage potential of an ecosystem, as different plants possess varying carbon sequestration capacities (*Zhang et al., 2017*). Therefore, understanding the intricacies of grazing management is paramount not just for ecosystem health, but also for optimizing the carbon sequestration potential of landscapes.

Recently, the difference in the effects of livestock grazing management on ecosystems has attracted much attention (*Eldridge & Delgado-Baquerizo, 2017*). Studies have reported on the effects of seasonal prohibited grazing and grazing intensities on soil organic carbon stock in Mongolian grasslands (*He et al., 2011*; *Chang et al., 2015*). Other management practices, including grazing period, number of burns, and slashes, have been considered in a dry tropical region of western Mexico (*Trilleras et al., 2015*). Although the effects of several grazing management practices have been reported separately in case studies, the role of grazing management in carbon stock, especially in vegetation, has not been fully compared.

Given the fragmented nature of current research on grazing management's effect on carbon stock, there's a pressing need for a more comprehensive approach to these investigations. The model simulation approach presents a robust avenue to address this gap. Through modeling, we can holistically evaluate the impacts of varied livestock grazing management practices against a unified benchmark, enabling more thorough comparisons. Firstly, models in plant population and vegetation dynamics have been widely reported, providing a framework for models (*Griffith & Forseth, 2005*; *Marano & Collalti, 2020*). Secondly, the quantitative relationship between carbon stock and plant biomass has been explored (*Bayen et al., 2020*; *Brown et al., 2020*). Lastly, experimental studies have provided some evidence to verify the simulating models (*Li, 2006*; *Zhao et al., 2016*).

Understanding the impacts of grazing management on vegetation carbon storage is not only of academic interest but also holds profound implications for policy formulation and the evaluation of ecosystem services. Firstly, optimizing carbon storage in ecosystems serves as a countermeasure to rising atmospheric $CO_2$ concentrations, potentially mitigating the impacts of global warming (*Piao et al., 2009*). Moreover, the retained carbon

stock is intrinsically linked to the broader spectrum of ecosystem services, underpinning their value and functionality (*Yang & Tang, 2019*). As grazing practices directly modulate the carbon balance of an ecosystem, they become pivotal tools in enhancing ecological services.

Carbon sequestration rates serve as a crucial aspect in understanding the role of grazing management in determining carbon fixation within vegetation. The carbon sequestration rate is an insightful metric to represent the annual carbon accumulation in vegetation (*Wang, Wang & Niu, 2014*; *He et al., 2017*). Calculating vegetation carbon sequestration rates is conventionally achieved by examining the average change in carbon stock over a designated time span (*Li et al., 2023*). In recent studies, it has been proposed that the relationship between vegetation carbon stocks and carbon sequestration rates can be effectively described using logistic functions (*Dong, Bettinger & Liu, 2021*). Utilizing logistic functions to determine carbon sequestration rates proves advantageous as it encompasses non-linear growth patterns, offering a comprehensive perspective on carbon accumulation dynamics over time.

The primary objective of this study was to assess the impacts of livestock grazing management on vegetation in semi-arid regions from two perspectives: carbon stock and carbon sequestration rates. Our approach involved establishing a system dynamics model with multiple life-history stages to simulate the effects of grazing managements on plant biomass. By understanding the relationship between biomass and carbon stock, we could determine the carbon stock. Subsequently, by employing a logistic model linking carbon stock to carbon sequestration rates, we estimated the latter. This research provides a holistic methodology to comprehend the multifaceted repercussions of grazing management on carbon dynamics, thereby offering valuable insights for sustainable land-use strategies in semi-arid environments.

# MATERIALS AND METHODS

This study was conducted in the Wulanaodu region (42° 29′–43° 06′N, 119° 39′–120° 02′E, 480 m a.s.l.), located in the Horqin Sandy Land, one of the largest sand lands in China (*Tang, Jiang & Lv, 2014*). This region is a typical landscape composed of active sand dunes, stabilized sand dunes, and inter-dune lowlands. The region has a semi-arid climate, with a mean temperature of 23 °C in July and –14 °C in January (*Zhang et al., 2016*). Sparse elm woodlands are the original vegetation community in this region, and *Ulmus pumila* is the dominant tree species in these woodlands (*Tang, 2020*). Besides *U. pumila* trees, there are also sparsely distributed shrubs and herbs, whose biomass is far less than *U. pumila* trees (*Zhao et al., 2016*).

## System dynamic model

A system dynamic model was constructed, taking into account the five stages of the *U. pumila* life cycle, *i.e.*, seed (i = 1), seedling, juvenile, mature, and over-mature (i = 5) tree stages. In this model, the seed stage is linked to the mature and over-mature stages, as seeds are produced in these two stages. The seeds become seedlings at a specific germination rate. *U. pumila* trees die at a stage-specific rate, except for the seed stage. At the over-mature

**Table 1 Values of parameters used in the system dynamic model.**

| Parameters | Seed | Seedling | Juvenile tree | Mature tree | Over-mature tree |
|---|---|---|---|---|---|
| Seed production (seeds·m$^{-2}$) | – | – | – | 2,456 | 7,744 |
| Death rate (%) | – | 0.268 | 0.133 | 0.346 | 0.941 |
| Expectation in precipitation (mm) | 352 | 352 | 352 | 352 | 352 |
| Variance in precipitation | 8,627 | 8,627 | 8,627 | 8,627 | 8,627 |
| Seed germinatied rate (%) | – | – | – | 10 | 10 |
| Period of stages (years) | 1 | 5 | 15 | 30 | 50 |
| Transition probability (%) | – | 20 | 6.67 | 3.33 | 2 |
| Water consumption (L/individuals year) | – | 21.6 | 1,680.7 | 1,977 | 1,977 |
| RG5 | 1.51 | 0.56 | 0.89 | – | – |
| CG | 1.51 | 0.56 | 0.89 | – | – |
| SPG | – | 0.56 | 0.89 | – | – |

stage, no transition is made to the next stage; as a result, the over-mature stage is the last in the *U. pumila* life cycle. The equations depicting the elm life cycle are shown below (*Tang & Busso, 2018*).

$$\frac{dN_i}{dt} = \sum_{i=4}^{5} B_i \cdot N_i - TP_i \quad \cdots\cdots\cdots\cdots\cdots\cdots\cdots\cdots \quad \textit{for seed stage} \tag{1}$$

$$\frac{dN_i}{dt} = \left\{ \begin{array}{ll} TP_{i-1} \cdot N_{i-1} - (D_i + TP_i) \cdot N_i, N_i < M_i \\ 0 \qquad\qquad\qquad\qquad , N_i \geq M_i \end{array} \right\}, i = 2, \cdots, n-1 \tag{2}$$

$$\frac{dN_i}{dt} = \left\{ \begin{array}{ll} TP_{i-1} \cdot N_{i-1} - D_i \cdot N_i, & N_i < M_i \\ 0 & , N_i \geq M_i \end{array} \right\} \cdots\cdots\cdots\cdots i = n \tag{3}$$

$$M_i = \frac{P_i}{WC_i}, \cdots\cdots\cdots\cdots\cdots\cdots\cdots\cdots\cdots\cdots\cdots\cdots i = 2, \cdots, n \tag{4}$$

Here, N represents the number of individuals at each of the five stages, while B and D represent birth and death rates, respectively. TP stands for the transition probability (*i.e.*, the probability associated with an elm population at one age-stage shifting to another age-stage). WC represents the water consumption per individual at each of the developmental stages, except for the seed stage. P represents precipitation, which follows a normal distribution, whose parameters, *i.e.*, mean and variance, were calculated according to long-term collected data. Besides the parameters mentioned above, other parameters include death rate, seed germination rate, and the period of each morphological stage. The values of all parameters used in the system dynamic model are shown in Table 1.

## Carbon stock

The carbon stock of elm trees was calculated using Eq. (5), where the C content of elm trees was calculated by considering different parts of the elm trees, including leaves, twigs, stems, and roots.

**Table 2 The values of parameters describing the allometric growth and carbon contents of leaf, twig, stem, and root in elm trees.**

| Parameters | Leaf | Swig | Stem | Root |
|---|---|---|---|---|
| a | 0.033 | 0.0303 | 0.0146 | 0.0146 |
| b | 1.7241 | 2.3445 | 2.5837 | 2.893 |
| C content (g Kg$^{-1}$) | 406 | 444 | 446 | 424 |

$$Total\ C = \sum_{i=1}^{4} C\ concentration_i \times Biomass_i \tag{5}$$

In Eq. (5), i ranges from 1 to 4, representing the leaf, twig, stem, and root separately. The C concention of each part of the elm trees was specifically measured in a previous study (Zhao et al., 2016, Table 2A). The biomass in each part was calculated using allometric equations, the formation of which follows Eq. (6). Species-specific allometric equations based on tree diameter at breast height (DBH) were applied to estimate the biomass.

$$Biomass_i = a_i \times DBH^{b_i} \tag{6}$$

where $a_i$ and $b_i$ are special parameters measured in a previous study (Li, 2006, Table 2), i from 1 to 4 representing leaf, twig, stem, and root separately. DBH was estimated using Eq. (7), with the age of the elm tree serving as the independent variable.

$$DBH = A + B \times age \tag{7}$$

where A and B are estimated using data from a previous study and their values here are −0.1698 and 0.3839 (Niu, 2008).

## Carbon sequestration rate

The vegetation carbon sequestration rate (Rseq) was obtained with a logistic model, where the carbon sequestration rate serves as a parameter and is estimated using Eq. (8).

$$Carbon\ stock = K \times N_0 \times \frac{e^{R_{seq} \times age}}{K + N_0 \times (e^{R_{seq} \times age} - 1)} \tag{8}$$

In Eq. (8), carbon stock refers to the amount of carbon stored in the vegetation at a given age. K is the carrying capacity, representing the maximum potential carbon stock achievable by the vegetation in the given conditions. $N_0$ is the initial carbon stock at age 0, representing the amount of carbon present in the vegetation at the time of planting or the start of observation. $R_{seq}$ is the C sequestration rate, which governs the rate at which the vegetation assimilates atmospheric carbon. Here K, $N_0$ and $R_{seq}$ are parameters needed to estimate.

## Scenarios analysis

We evaluated the effects of grazing management on carbon stock in sparse elm woodlands. In this study, we considered four scenarios of grazing management, i.e., rotational grazing

every 5 years (RG5), prohibited grazing (PG), seasonal prohibited grazing (prohibition in periods from March to July, SPG), and continuous grazing (CG). Continuous grazing increases elm seed production and decreases elm seedling densities in sparse elm woodlands in a previous study (*Tang, Jiang & Lv, 2014*). Seasonal prohibited grazing influences elm seedlings but does not influence seed production, as seed dispersal mainly occurs in May (*Liu & Tang, 2018*). Meanwhile, rotational grazing every 5 years works as continuous grazing once every 5 years. The specific values are shown in Table 1.

## Data analysis

The system dynamic model was formulated and simulated using professional SD software (Vensim PLE). A unit-consistency test, which checks for agreement among units, was used to validate this model. It was automatically completed in the Vensim package (*Tang & Li, 2018*).

The models in Eqs. (7) and (8) were estimated using the R programming language (*R Core Team, 2022*). The best-fit model derived from Eq. (7) is $DBH = -0.169 + 0.383 \times age$ (F-statistics = 431.8, adjusted R2 = 0.837, $P < 0.05$). The parameters were significant at a critical value ($P < 0.05$). The standard error (SE) of the estimated parameters and residual standard error (RSE) of the logistical model were reported.

# RESULTS

## Effects of grazing management on carbon stock

In the first 10 years, the carbon stock of elm trees was almost the same under all four management techniques. From the 10th year to a 100 years, the carbon stock of elm trees in RG5 and PG management was higher than in CG and SPG management. From the 20th year to a 100 years, the carbon stock of elm trees under CG management was more substantial than under SPG management.

When comparing RG5 to PG management, the carbon accumulation in the PG treatment surpassed that of the RG-5 between the 70th and 80th years, but was diminished between the 90th and 100th years. In contrast, when comparing RG5 with CG management, carbon sequestration in CG was inferior to that in RG-5.

From the beginning to the 80th year, the ability to store carbon increased under RG5, PG, CG, and SPG management. After the peak, the ability to maintain carbon stock decreased under RG5, PG, and CG management, while the ability to maintain carbon stock remained steady under SPG management (Fig. 1).

The carbon stock of elm trees remained below 5 M g ha$^{-1}$ throughout the simulation period under SPG management. In contrast, the carbon stock of elm trees exceeded 5 M g ha$^{-1}$ almost from the 15th year and onwards under CG management. Additionally, the carbon stock of elm trees exceeded 5 M g ha$^{-1}$ after the 20th year under RG5 and PG management. Furthermore, the carbon stock of elm trees under RG5 and PG management ranged between 10 and 15 M g haz$^{-1}$ after the 40th year (Fig. 1).

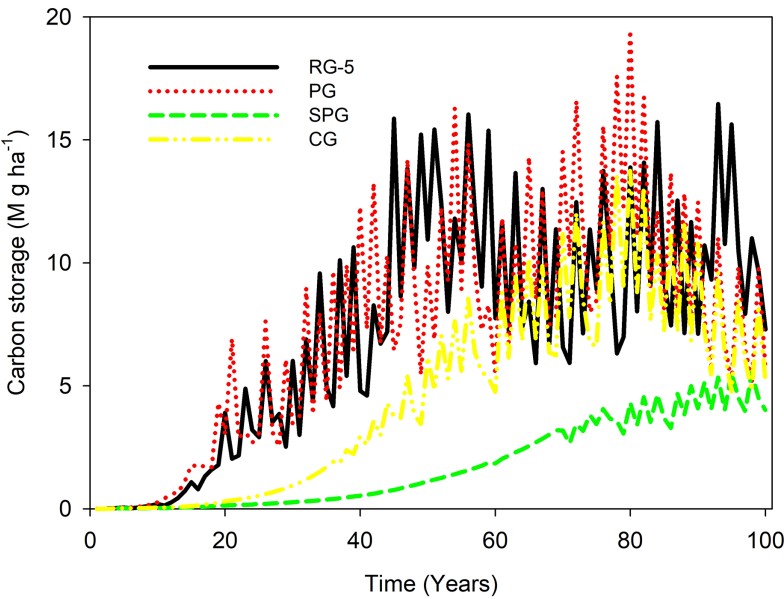

**Figure 1 Carbon stock in *U. pumila* of sparse woodlands under livestock grazing management.**

**Table 3 The values of estimated parameters in logistical models of carbon stock.**

| Scenarios | K | | $N_0$ | | $R_{seq}$ | | RSE |
|---|---|---|---|---|---|---|---|
| | Mean | SE | Mean | SE | Mean | SE | |
| RG5 | 10.5 | 0.39 | 0.1 | 0.11 | 0.15 | 0.03 | 2.635 |
| PG | 10.43 | 0.45 | 0.24 | 0.21 | 0.13 | 0.03 | 2.914 |
| SPG | 4.9 | 0.15 | 0.02 | 0.01 | 0.09 | 0.01 | 0.3071 |
| CG | 8.65 | 0.32 | 0.02 | 0.02 | 0.13 | 0.02 | 1.648 |

### Effects of grazing management on carbon sequestration rate

The carbon sequestration rate was significantly related to the carbon stock in elm trees under all four management techniques. The estimated values of the carbon sequestration rate in elm trees were 0.15, 0.13, 0.13, and 0.09 Mg C ha$^{-1}$ year$^{-1}$ in RG5, PG, CG, and SPG management, respectively. Additionally, under all four management techniques, the K values were also found to be significantly related to the carbon stock in elm trees (Table 3, Fig. 2).

## DISCUSSION

### Carbon stock in elm trees

In RG5 and PG management, the carbon stock in elm trees is faster and in greater quantities than in CG and SPG management. This suggests that RG5 and PG are the preferred choices for maintaining steady carbon storage in elm trees. This result is consistent with a previous study, where elm seedlings and sapling densities were found to be significantly higher in fenced plots than in grazed plots (*Tang, Jiang & Lv, 2014*). This is

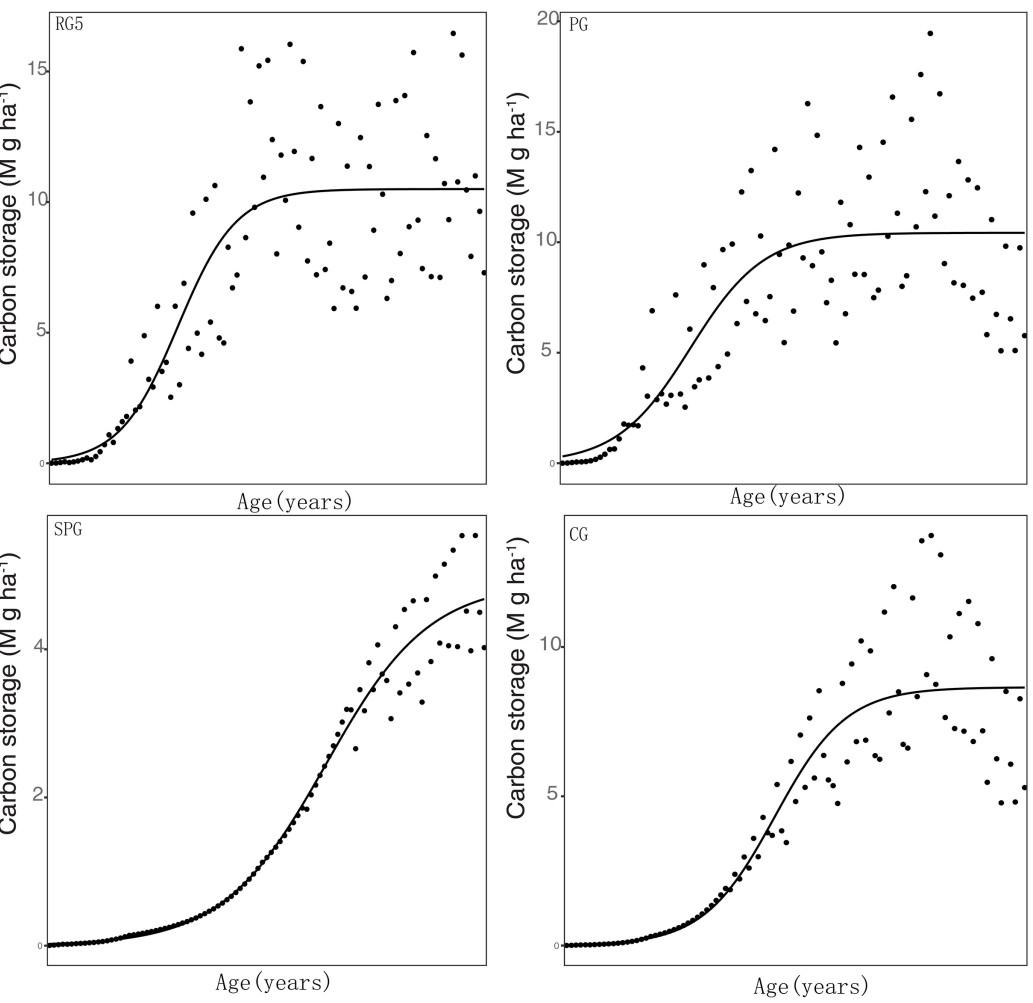

**Figure 2 Simulation results of carbon stock were fitted with logistical models.**

likely due to grazing threats to the growth of seedlings and saplings (*Bergmeier, Petermann & Schroeder, 2010*). Similar results were also found in a similar landscape, the tropical savanna, where long-term grazing was found to have reduced vegetation biomass (*Ngatia et al., 2015*).

According to this study, the carbon stock of elm trees is 10–15 M g ha$^{-1}$ in mature-stage sparse woodlands under prohibited grazing management. This result is consistent with an experimental study in sparse woodlands, where the carbon stock of elm trees was found to be 15.57 M g ha$^{-1}$ in fenced plots (*Zhao et al., 2016*). These consistent results suggest that the system dynamic approach is effective in simulating changes in the carbon stock of plant populations. In previous studies, the system dynamic approach has been used to simulate various aspects of plant populations, including the carbon cycle, hydrological processes, and population dynamics (*Mukherjee, Ray & Ghosh, 2013*; *Ouyang et al., 2016*; *Tang, Liu & Wu, 2016*). This study expands the application of the system dynamic

approach by incorporating it in the calculation of vegetation carbon sequestration and evaluating the effects of specific factors.

The results reveal a notably low carbon sequestration associated with the SPG treatment (Fig. 1). A plausible rationale for this observation hinges on the period spanning March to July, which is likely the predominant growth phase for regional vegetation. By prohibiting grazing during these crucial months, an excessive proliferation of herbs may ensue, which could subsequently impede the germination and establishment of elm tree seeds (Tang, Jiang & Lv, 2014). Such a reduction in the available seed source inevitably results in a decline in the population of elm trees. This chain reaction, commencing from the lowered elm population, can be linked to the observed decrement in vegetation carbon sequestration within the SPG treatment.

Meanwhile, the carbon accumulation observed in the PG treatment appears to surpass that of the RG-5 treatment between 70 and 80 years. This can potentially be attributed to the enclosure practices, which tend to bolster biomass accumulation, yielding a slight elevation in carbon sequestration compared to RG5 within that timeframe. Yet, as the duration of enclosure prolongs, an elevated population density ensues, inciting heightened interspecific competition. This amplified competition paves the way for increased individual mortality, leading to a reduction in the aggregate population biomass, and, in turn, a subsequent decline in carbon accumulation.

Grazing management can influence soil carbon stock by regulating soil organic carbon and soil microbial carbon (Li et al., 2013). The effects of grazing exclusion on soil carbon stock are more commonly reported compared to other grazing management techniques (Li et al., 2012). A possible reason for this is that grazing and grazing exclusion are the most commonly used grazing management techniques in grasslands and degraded lands, and it is relatively easier to establish designed experimental plots for these methods. Rotational grazing management has great potential for maintaining vegetation carbon stock, and its effects on soil carbon stock should be fully considered.

## Carbon sequestration rate under grazing managements

Elm trees exhibited varying carbon sequestration rates under different management practices: 0.15, 0.13, 0.13 and 0.09 Mg C ha$^{-1}$ year$^{-1}$ in RG5, PG, CG, and SPG, respectively. This clearly highlights RG5 management as the most efficient in terms of carbon sequestration, while SPG management lagged behind the others. Although both CG and PG management exhibited the same carbon sequestration rate, the carrying capacity (K value) was notably higher under PG management. This could account for the observed higher carbon stock in PG as compared to CG throughout the duration of the study. This data suggests a hierarchical efficiency in carbon sequestration by elm trees under different management practices: RG5 being the most efficient, followed by PG, CG, and then SPG.

Management practices can significantly shape vegetation carbon sequestration rates by influencing the structural and functional attributes of plant communities (Whittinghill et al., 2014). For instance, practices that promote plant diversity and maintain a dense vegetative cover tend to enhance the overall photosynthetic capacity of an ecosystem

(*Quijas, Schmid & Balvanera, 2010*). A diverse plant community, with species that have varied photosynthetic rates and growth patterns, can capture carbon more efficiently throughout the year (*Zhang, Wu & Tang, 2016*). Moreover, management interventions that optimize nutrient availability, such as controlled grazing or periodic soil amendments, can boost plant growth and thereby enhance their carbon intake (*Ma et al., 2019*).

Conversely, mismanaged practices can have deleterious effects on vegetation carbon sequestration. Overgrazing, for example, can result in the dominance of less productive plant species, diminishing the overall carbon capture capacity of the vegetation (*Senbeta et al., 2013*). Practices that lead to soil compaction or erosion can hinder root growth, thereby reducing the plant's ability to access water and nutrients essential for photosynthesis (*Kim et al., 2010*). Additionally, the loss or suppression of certain plant species due to specific management choices can disrupt the synchrony between plant phenology and climatic patterns, potentially leading to decreased photosynthetic periods (*Pathare et al., 2017*). Hence, tailoring management practices to the ecological needs of the vegetation is vital to maximize carbon sequestration rates.

Our findings, indicating carbon sequestration rates for elm trees ranging between 0.09 and 0.15 under various management measures, align with those reported by *Bhatta, Sharma & Balami (2018)* in Central Nepal. *Bhatta, Sharma & Balami (2018)* documented a carbon sequestration rate of 0.14 Mg C ha$^{-1}$ year$^{-1}$ for trees within the Ulmaceae family. This further attests to the appropriateness of employing logistic models to capture the accumulation dynamics of biomass in vegetation. It is worth noting, however, that the carbon sequestration rates derived from our logistic models represent an average value across the entire life cycle of the vegetation and do not imply that the annual carbon sequestration rates for the plants consistently maintain this state.

## Other factors influencing carbon stock in semi-arid lands

In addition to grazing management, the duration of grazing and grazing exclusion management can also influence carbon storage in soil and vegetation (*Gebregergs et al., 2019*). For example, ecosystem carbon storage in an alpine meadow steppe showed a hump-shaped pattern in response to the duration of grazing exclusion with a 6-year threshold (*Li et al., 2018*). In this study, the results provide evidence that the duration of grazing management influences vegetation carbon storage in sparse woodlands. Moreover, longer periods of grazing exclusion management do not necessarily lead to more vegetation carbon storage in over-mature trees. While our study emphasizes the importance of considering the duration of grazing management in decision-making processes to enhance carbon sequestration, it is pivotal to also scrutinize the specific strategies implemented during grazing. This encompasses analyzing the length and timing of rotations, as the efficacy of grazing management on carbon storage in sparse woodlands might be intricately linked to these variables. Moreover, integrating auxiliary strategies such as selective cutting, especially when trees exhibit diminished carbon storage capabilities, could serve as a pragmatic approach to sustain and potentially augment carbon sequestration in these ecosystems (*Hulvey et al., 2013*). Thus, a holistic approach, examining both the temporal and strategic facets of grazing management alongside
supplementary woodland management practices like selective cutting, should be adopted to cultivate a robust framework that optimally supports carbon sequestration.

Climate changes, especially changes in precipitation and temperature, can affect vegetation carbon stock as biomass accumulation is influenced (*Dai et al., 2013*). Precipitation plays a vital role in regulating plant growth and vegetation patterns in terrestrial ecosystems (*Peng et al., 2017*). In sparse elm woodlands, population densities, but not the population age structure of elm trees, are greatly influenced by precipitation (*Tang & Busso, 2018*). The effects of precipitation on carbon storage in sparse elm woodlands are worth exploring in future research.

Livestock grazing can decrease the herb layer in terms of biomass, coverage, and spatial distribution (*Eldridge et al., 2020*; *Trigo et al., 2020*). Subsequently, the changes in herbs might affect the growth of elm trees, especially in the early stages of their life history. However, the effects of herbs on elm tree growth are relatively difficult to evaluate. Firstly, herbs play different roles in influencing the secondary dispersal of elm seeds. Herbs with vegetation coverage lower than the threshold can promote seed secondary dispersal, while herbs with vegetation coverage higher than the threshold can impede seed secondary dispersal (*Jiang, Tang & Busso, 2014*). Secondly, herbs have little effect on elm seed germination and seedling growth, as their effects might be mixed with other factors, such as light and litter biomass (*Vaz et al., 2019*). Thirdly, microbiomes might connect the interaction between herbs and elm trees, yet the role of microbiomes on plants in sparse elm woodlands is only recently being considered (*Liang et al., 2019*). Therefore, considering the interaction between herb layers and elm trees could promote the accuracy of tree growth models.

## CONCLUSIONS

In our research, we crafted an advanced system dynamics model that integrates multiple life-history stages to explore the impact of livestock grazing management strategies on carbon storage, both in terms of carbon stock and sequestration rates. Importantly, for a nuanced estimation of carbon sequestration rates, we employed logistical models.

In this study, we examined four distinct grazing management strategies: RG5, PG, CG, and SPG. Our findings indicate that the carbon stock of elm trees under RG5 and PG management surpassed those observed under CG and SPG management. Moreover, our research highlights the differences in carbon sequestration rates of elm trees across these managements, with RG5 standing out as the most effective. As a result, our data suggests that rotational grazing management may offer a superior strategy to enhance vegetation carbon storage in sparse elm woodlands in semi-arid landscapes. Furthermore, in line with previous research, our logistic model outcomes showed that the carbon sequestration rates for elm trees ranged between 0.09 and 0.15 Mg C ha$^{-1}$ year$^{-1}$.

Our assessment sheds light on the carbon sequestration capabilities of vegetation in sparse elm woodlands, offering valuable insights for grazing management in semi-arid regions. For a more refined understanding of carbon storage capacities in the future, it will be imperative to integrate soil carbon sequestration studies and delve deeper into the interactions between the herb layers and elm trees within these sparse woodlands.

### Funding

This work was supported by the National Natural Science Foundation of China (31870709) and the Scientific Research Funding Project of the Education Department of Liaoning Province (LJKZ0103). The funders had no role in study design, data collection and analysis, decision to publish, or preparation of the manuscript.

### Grant Disclosures

The following grant information was disclosed by the authors:
National Natural Science Foundation of China: 31870709.
Scientific Research Funding Project of the Education Department of Liaoning Province: LJKZ0103.

### Competing Interests

The authors declare that they have no competing interests.

### Author Contributions

- Yi Tang conceived and designed the experiments, performed the experiments, analyzed the data, prepared figures and/or tables, authored or reviewed drafts of the article, and approved the final draft.

### Data Availability

The raw data is available in Tables 1–3.

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
