# Peer review of "Impact of livestock grazing management on carbon stocks: a case study in sparse elm woodlands of semi-arid lands"

_PeerJ, doi:10.7717/peerj.16629_

## Round 0.1 · original submission · Major Revisions

Based on the reviewers' recommendations, my decision is that the manuscript needs major revisions. In addition to the reviewers' comments, please take into account my own comments, included in the annotated manuscript.

Above all, the discussion needs to be improved. There is no explanation of the results of this study. Particularly, the abnormally low carbon sequestration in treatment SPG deserves an interpretation.

Moreover, in the long term (60-90 yrs) carbon sequestration in GG is very similar to that in RG-5. Carbon accumulation in continuous grazing is low in the short term (less than 60 years), although always higher than in the SPG treatment. From 60 years of age, the accumulation of C in the CG treatment approaches that of the RG-5 treatment. Why?

Regarding the PG treatment, it produces a C accumulation similar to the RG-5 treatment up to approximately 65 years. C accumulation in the PG treatment is greater than that of the RG-5 treatment between 70 and 80 years and lower between 90 and 100 years. Why?

**Language Note:** PeerJ staff have identified that the English language needs to be improved. When you prepare your next revision, please either (i) have a colleague who is proficient in English and familiar with the subject matter review your manuscript, or (ii) contact a professional editing service to review your manuscript. PeerJ can provide language editing services - you can contact us at copyediting@peerj.com for pricing (be sure to provide your manuscript number and title). – PeerJ Staff

·

Basic reporting

This ms is written with prosessional atructure and english, sufficient field background provided.

Experimental design

The study stands out as original and crucial. It fills a significant knowledge gap in understanding the effects of livestock grazing management on carbon sequestration, particularly in sparse elm woodlands.

Validity of the findings

The study results, showing different carbon sequestration rates under various grazing management strategies, are intriguing and significant. However, I recommend that the authors expand on the differences observed and offer possible explanations for the effectiveness of rotational and prohibited grazing over other strategies. It would also be insightful if the authors discuss whether these findings are generalizable to other types of vegetation or geographic locations.

Additional comments

The study stands out as original and crucial. It fills a significant knowledge gap in understanding the effects of livestock grazing management on carbon sequestration, particularly in sparse elm woodlands.

The study results, showing different carbon sequestration rates under various grazing management strategies, are intriguing and significant. However, I recommend that the authors expand on the differences observed and offer possible explanations for the effectiveness of rotational and prohibited grazing over other strategies. It would also be insightful if the authors discuss whether these findings are generalizable to other types of vegetation or geographic locations.

The research has significant practical implications, especially for policy makers and stakeholders involved in livestock grazing management. I suggest that the authors include some suggestions for how these findings might be incorporated into policy or management practices.

Overall, the manuscript has a good scientific basis, but it needs a few revisions to improve readability and clarity.

Specific comments:
"Therefore, increasing vegetation carbon storage is an effective way to mitigate global warming." --> "Therefore, strategies that enhance vegetation carbon storage can effectively mitigate global warming."

"The terrestrial biosphere, which holds approximately 2,000 Gt of carbon, plays a crucial role in carbon storage (Ruhland and Niere,2019)." --> "The terrestrial biosphere, a repository of approximately 2,000 Gt of carbon, plays a crucial role in carbon storage (Ruhland and Niere,2019)."

"In these areas, livestock grazing is a significant type of human activity." --> "In these regions, livestock grazing is a prominent human activity."

"A system dynamic model was built considering five stages of the U. pumila life cycle..." --> "A system dynamic model was constructed, taking into account the five stages of the U. pumila life cycle..."

"Where N represents the number of individuals at each of the five stages, B and D represent the birth and death rates, and TP represents the transition probability" --> "Here, N represents the number of individuals at each of the five stages, while B and D represent birth and death rates, respectively. TP stands for the transition probability..."

Please consider using a more descriptive variable name instead of "Max" in equations (2), (3), and (4). This will make your mathematical representation more understandable.

"The DBH was estimated with equation 7, where the age of elm tree serves as an independent variable." --> "DBH was estimated using equation 7, with the age of the elm tree serving as the independent variable."

"The estimated linear models in equations 7 and 8 were calculated using R programming language (R Core Team, 2022)." --> "The linear models in equations 7 and 8 were estimated using the R programming language (R Core Team, 2022)."

"The fittest model in equation 7 is DBH=-0.169+0.383×age (F-statistics= 431.8, adjusted R2= 0.837, P <0.05)." --> "The best-fit model derived from equation 7 is DBH=-0.169+0.383×age (F-statistics= 431.8, adjusted R2= 0.837, P <0.05)."

These suggestions should enhance the clarity and readability of your manuscript. Please also ensure that all your references are correctly cited and formatted according to the required style.


"This might be due to grazing threats to the growth of seedlings and saplings (Bergmeier et al.,2010)." - 'Might be' implies uncertainty. If this is a proven fact, consider revising to 'This is likely due to...'

"In this study, the carbon sequestration rate was calculated using logistic models." - This information may be better placed in a methods section.


"Rotational grazing and prohibited grazing management perform well in increasing vegetation carbon storage in sparse woodlands." - It would be beneficial to quantify how well these management practices performed in comparison to others.


"The carbon stock of elm trees fell within the range of 10 M g ha-1 to 15 M g ha-1 after the 40th year under RG5 and PG management." - Consider revising to "The carbon stock of elm trees under RG5 and PG management ranged between 10 and 15 M g ha-1 after the 40th year."

"The value of the carbon sequestration rate was the highest in PG5 management and the lowest in SPG management." - Consider revising to "The carbon sequestration rate was highest under PG5 management and lowest under SPG management."

·

Basic reporting

Dear author, I have reviewed the paper "Carbon stocks of sparse forests are regulated by the livestock grazing management in semi-arid lands", which aimed to compare the effects of cattle grazing management on forest carbon stocks... and to provide a model to estimate carbon sequestration.

The study is generally well written. Regarding the design and structure of the scientific technical content, I consider that the author should order and be clear about how quantifiable and presentable the objectives can be. The idea of the methodological process is to present it in a clear way so that it can be easily replicated. The results are very sketchy and differ from the stated objectives. The discussion should be further strengthened with other studies and mark those similarities and differences. In the conclusions, the author should present those ideas that respond to the contrast of the objectives and their results.

 In the second half of the summary section the author should group the results and then conclude. In the current form there is a mixture of results and conclusions.
 From line 81 onwards, the author should describe specific data on carbon sequestration, the wording is very superficial. It would be important to specify quantities and evolution over time. This will allow a clear idea to approach the discussion after the results of this study.
 Prior to the last paragraph of the introduction, it is necessary to describe the problem and justify the research, this will allow connecting with the final recommendations presented in the conclusions.

Experimental design

I ask the author: This methodological model was applied in other studies, to give robustness to its metrology, it is necessary to cite other studies where part or all of the methodological process used in this study was applied.

 In the discussion you should generate two very well marked sections that respond to each of your objectives and/or research questions.

 The conclusions should respond to each of your objectives and/or research questions. To make it easier to organize your study I recommend in this section to generate a paragraph for each objective/question. Additionally, you need to generate a final paragraph on the limitations of your study and future studies based on your results.

Validity of the findings

The author should be orderly in presenting the results. I suggest you review the objectives presented in the last paragraph of the introduction, these are two: To compare the effects... and to provide a model... Where is it? If you want to provide a model, do you present this or is this what is in the methodology?

Reviewer 3 ·

Basic reporting

The authors present a study on Carbon stocks of sparse forests are regulated by the livestock grazing management in semi-arid lands. The study is interesting, however, the paper is not well written. The novelty of the study is not indicated compared to previous study.

The title seems like a hypothesis. It is written as the result of the work is already known. So, I recommend revision of the title.

The introduction part needs some revision. How does grazing management regulate carbon stock? The mechanisms should be clearly indicated in the introduction part. What does it mean sparse forests in the context of the study area? How are the sparse forests managed in the study area? The author should provide a full picture about the title understudy otherwise the introduction is very shallow. The flow of the paragraphs and the language should be improved.
Specific corrections and questions that needs clarity:
Line 39: CO2 should be replaced by CO2 throughout the manuscript.
Line 48: provide reference for the statement.
Line 76: “management practices that maintain more vegetation carbon storage are preferred”. How do you reconcile this statement with the interest of the community?
Line 78: “Grazing management can influence ecological services by regulating carbon stock in ecosystems.” How does grazing management regulate carbon stocks?

Experimental design

Does the sparse forest only contains elm trees? The site description, study design and sampling method should be clearly indicated in the materials and methods? The study design should come before the carbon stock analysis. How was the samples taken? Over all the methodology is not clearly written. Needs through revision.
What are the special parameters in equation 6?
What are A and B in equation 7?
What are K and N0 in equation 8? Mention the parameters?
Does the allometric equation used in this study was developed by the author?

Validity of the findings

Findings are clearly written in the manuscript. However, there are things that needs clarity.
How the values of carbon stocks are calculated?
Are the values of carbon sequestration rate per year?
What are a, b, c, d indicate in Figure 2.

Additional comments

In the conclusion part, the author recommended logistical models should be used instead of linear models to evaluate the carbon sequestration rate in vegetation. How was that evaluated?

---

## Round 0.2 · accepted · Accept

The authors have undertaken commendable efforts in substantially enhancing the article, taking into consideration both the recommendations provided by the reviewers and my own suggestions. The manuscript is now poised for publication in PeerJ. Congratulations!

·

Basic reporting

This ms has been revised according to my comment and can be accept as it is.

Experimental design

The experimental design has improved greatly and achieved the publication standards.

Validity of the findings

The findings has important significance on livestock grazing management in semiarid grasslands.

Additional comments

No comment.